# Cloning and functional characterization of seed-specific *LEC1A* promoter from peanut (*Arachis hypogaea* L.)

**Guiying Tang[1], Pingli Xu[1], Pengxiang Li[1,2], Jieqiong Zhu[1,2], Guangxia Chen[3], Lei Shan[1,2]\*, Shubo Wan[1,2]\***

**1** Bio-Tech Research Center, Shandong Academy of Agricultural Sciences / Shandong Provincial Key Laboratory of Crop Genetic Improvement, Ecology and Physiology, Jinan, Shandong, China, **2** College of Life Science, Shandong Normal University, Jinan, Shandong, China, **3** Shandong Academy of Grape, Jinan, Shandong, China

\* shlei1025@sina.com (LS); wansb@saas.ac.cn (SW)

**Data Availability Statement:** All relevant data are within the manuscript.

**Funding:** Our work was funded by the National Key R&D Program of China (2018YFD1000906), and

## Abstract

LEAFY COTYLEDON1 (LEC1) is a HAP3 subunit of CCAAT-binding transcription factor, which controls several aspects of embryo and postembryo development, including embryo morphogenesis, storage reserve accumulation and skotomorphogenesis. Herein, using the method of chromosomal walking, a 2707bp upstream sequence from the ATG initiation codon site of *AhLEC1A* which is a homolog of Arabidopsis *LEC1* was isolated in peanut. Its transcriptional start site confirmed by 5' RACE was located at 82 nt from 5' upstream of ATG. The bioinformatics analysis revealed that there existed many tissue-specific elements and light responsive motifs in its promoter. To identify the functional region of the *AhLEC1A* promoter, seven plant expression vectors expressing the *GUS* (β-glucuronidase) gene, driven by 5' terminal series deleted fragments of *AhLEC1A* promoter, were constructed and transformed into Arabidopsis. Results of GUS histochemical staining showed that the regulatory region containing 82bp of 5' UTR and 2228bp promoter could facilitate GUS to express preferentially in the embryos at different development periods of Arabidopsis. Taken together, it was inferred that the expression of *AhLEC1A* during seed development of peanut might be controlled positively by several seed-specific regulatory elements, as well as negatively by some other regulatory elements inhibiting its expression in other organs. Moreover, the *GUS* expression pattern of transgenic seedlings in darkness and in light was relevant to the light-responsive elements scattered in *AhLEC1A* promoter segment, implying that these light-responsive elements harbored in the *AhLEC1A* promoter regulate skotomorphogenesis of peanut seeds, and *AhLEC1A* expression was inhibited after the germinated seedlings were transferred from darkness to light.

## Introduction

Seed development is a complex procedure of the flowering plant in life cycle, which can conceptually be divided into two distinct phases: embryo morphogenesis and seed maturation

the Major Basic Research Project of Natural Science Foundation of Shandong Province (2018GHZ007). In addition, the funders had no role in study design, data collection and analysis, decision to publish, or preparation of the manuscript.

**Competing interests:** The authors have declared that no competing interests exist.

[1, 2]. Lots of genes highly and specifically expressed in different developmental processes highlight the importance of transcriptional regulations for proper seed formation [3]. The Arabidopsis *LAFL* genes coding for LEAFY COTYLEDON1 (LEC1), ABSCISIC ACID INSENSITIVE3 (ABI3), FUSCA3 (FUS3), and LEC2, respectively form a network involved in regulating seeds development [4–7].

LEC1 is a central regulator controlling embryogenesis and seed maturation in *Arabidopsis thaliana* [8–10]. It expresses primarily in the embryo and endosperm, particularly during seed development [9, 11]. Loss of *LEC1* function causes a pleiotropic phenotype, including cotyledon with trichomes, distortions in suspensor-cell specification, defects in storage protein and lipid accumulation, embryo desiccation-intolerance in developing seeds, and leaf primordia initiation [8, 12, 13]. The expression of many genes involved with maturation processes are downregulated in *lec1* mutant seeds [9, 14, 15]. Moreover, the role of *LEC1* was also demonstrated by analyzing its gain-of-function mutant. For example, overexpression of *LEC1* in developing seeds elevates the contents of seed storage macromolecule, as well, upregulates the key genes involved in storage protein and lipid accumulation in a number of plant species [16–20]. Genome-wide analysis of *LEC1* occupancy and interactome indicated that LEC1 regulate many genes involved in embryo development [14, 15]. Recently, several researches indicated that LEC1, a central regulator of seed development, interacts with different combinations of ABI3, bZIP67, FUS3, and other TFs to regulate diverse developmental processes at different stages of seed development [21, 22].

LEC1 also participates in regulating post-embryonic growth during the developmental transition from germination seeds to seedlings. In higher plants, rapid elongation of hypocotyl in germinating seeds can be induced by darkness. However, *lec1* mutant causes a reduced capability for hypocotyl elongation and apical hook formation [14, 23]. Additionally, the expression of *LEC1* was also detectable in etiolated seedlings [24, 25], and the phenotype of longer hypocotyls and higher expression levels of the genes involved in etiolated growth were observed in LEC1-overexpression plants [26]. A *LEC1* gain-of-function mutant turnip (*tnp*), displayed the partial de-etiolation at dark-grown condition, including inhibited hypocotyls elongation, and activated SAM (shoots apical meristem) [27].

Despite of such comprehensive knowledge about *LEC1* in the model plant *Arabidopsis*, much less is known about the expression patterns and functions in other higher plants. Here, the 2707bp 5' flanking region of peanut *AhLEC1A* was isolated, and the GUS expression profiles driven by a series of deletion in its 5' flanking region were characterized in transgenic Arabidopsis. The results showed that the regulatory region containing 82bp of 5' UTR and 2228bp promoter could specifically regulate *AhLEC1A* expressing in developing seeds. Thus, the *AhLEC1A* promoter could be utilized as a seed-preferential promoter for plant genetic engineering.

## Materials and methods

### Plant materials and growth conditions

Peanut (*Arachis hypogaea* L. cv. Luhua 14) seeds, *Arabidopsis thaliana* L. (Ecotypes Col) seeds, *Escherichia* coli strain DH5α, pCAMBIA3301 plasmid and *Agrobacterium tumefaciens* strain GV3101 used in the present study were maintained at our laboratory. Peanut plants were grown in the experimental field of Shandong Academy of Agricultural Sciences. Roots, stems and leaves of 14-day seedlings, flowers, and the developing seeds were collected and kept in -80°C refrigerator for isolation of total RNA.

**Table 1. List of primers used in the study.**

| Serial No. | Name | Sequence (from 5' to 3') | Feature |
|---|---|---|---|
| 1 | BD GenomeWalker adaptor | GTAATACGACTCACTATAGGGCACGCGTGGTCGACGGCCCGGGCTGGT | No.1-5 used for the amplification of 5' flanking sequence of *AhLEC1A* |
| 2 | LEC1AGSP1-2 | TCAACCCAGAGGTAGTGGTAGGAAGG | |
| 3 | LEC1AGSP2-2 | TGATAACCGTGAAAGCCTCCTCCAGT | |
| 4 | AP1 | GTAATACGACTCACTATAGGGC | |
| 5 | AP2 | ACTATAGGGCACGCGTGGT | |
| 6 | GeneRacer RNA Oligo | CGACUGGAGCACGAGGACACUGACAUGGACUGAAGGAGUAGAAA | No.6-10 used for the localization of the transcriptional start site of *AhLEC1A* gene |
| 7 | TSS LECGSP1-1 | TCTTTTGCGTCGTCGGAGATTTTAGC | |
| 8 | TSS LECGSP2-2 | TGATAACCGTGAAAGCCTCCTCCAGT | |
| 9 | 5' GeneRacer Primer | CGACTGGAGCACGAGGACACTGA | |
| 10 | 5' Nested Primer | GGACACTGACATGGACTGAAGGAGTA | |
| 11 | P1 | CCATGGGTGTGAAGAAAGATGCAGTG | No.11-18 used for the amplification of different length promoter fragments |
| 12 | P2 | AAGCTTCATTAGGGTCAAAAGAGTG | |
| 13 | P3 | AAGCTTTCTTGGCAATAAATGTTGG | |
| 14 | P4 | AAGCTTCCCGTTAAAAAAAATAATAAG | |
| 15 | P5 | AAGCTTGTAATTTTTGGATAGCTTG | |
| 16 | P6 | AAGCTTTACATGGCACGCCTCATATC | |
| 17 | P7 | AAGCTTAGATCGAAACTAATTTAAG | |
| 18 | P8 | AAGCTTAAAAAGTTGAACATTTTATATAG | |
| 19 | AhACTIN7-F | ATGTATGTAGCCATCCAAG | No.19-22 used for qRT- PCR analysis of *AhLEC1A* |
| 20 | AhACTIN7-R | ACCAGAGTCCAGAACAATA | |
| 21 | AhLEC1A-F | ATACTCATACAGATGATAAC | |
| 22 | AhLEC1A-R | TGTGGAACAAAAGCAGAAGT | |

## Cloning of the 5' flanking region of *AhLEC1A*

The peanut genomic DNA was isolated from Luhua 14 leaves using CTAB method [28]. Genome walking was performed to isolate the 5' flanking regulatory region. According to the BD Genome Walker Universal Kit (Clontech, USA) manufacturer's instructions, each of 2.5 μg genomic DNA was digested with four restriction enzyme *DraI*, *EcoRV*, *PvuII*, and *StuI* respectively; and then the digested samples were connected with the BD Genome-Walker adaptor resulting in the library containing digestions by *DraI*, *EcoRV*, *PvuII*, and *StuI* (LD, LE, LP, and LS). Based on the sequence of *AhLEC1A* genomic DNA, two nested gene-specific primers (GSP), LEC1AGSP1-2 and LEC1AGSP2-2, were designed. The first round of PCR reaction was done in a 25μL reaction system using an AP1 provided by Kit and LEC1A GSP1-2 as 5' terminal and 3' terminus primer, and 1μL DNA of each library as template. The nested PCR reaction was also performed using the same volume and conditions with primers AP2 and LEC1AGSP2-2, and 1μL of the 10-fold diluted primary PCR products as template. The specific PCR fragments from the second round reaction were isolated and inserted into the vector pEASY-T3. The recombinants harboring the target gene were validated by two-way sequencing using ABI3730 model DNA sequencer. The primer and adaptor sequences of this assay were listed in Table 1.

## Precise identification of transcription start site in *AhLEC1A*

The transcription start site of *AhLEC1A* gene was identified by 5' RACE (rapid amplification of cDNA ends) using a 5' RACE kit (Invitrogen GeneRacer™ Kit) following the instructions

provided by the manufacturer. Total RNA was extracted from the developing seeds of peanut Luhua 14 using the improved CTAB method [29]. The ds-cDNA was synthesized using the full-length mRNA with RNA Oligo as template. The ds-cDNA was cloned into vector pCR4-TOPO to establish the full-length cDNA library. According to the cDNA sequence of *AhLEC1A*, two 3' terminus gene-specific primers TSS LEC1AGSP1-1 and TSS LEC1AGSP2-2 were designed, for use in the nested PCR reaction. The 5' terminus general primer for two rounds of PCR were GeneRacer™ Primer and 5' Nested Primer. 1μL of the full-length cDNA library as got previously and a 50-fold dilution of the primary PCR product was used respectively as the template of the two rounds of PCR. The nested PCR products were collected and sequenced by ABI3730 model DNA sequencer. The primer sequences used in the assay were listed in Table 1.

## Expression analysis of *AhLEC1A* gene in various organs

The expression analysis was performed by qRT-PCR using ABI 7500 instrument. Gene-specific primers were designed according to *AhLEC1A* cDNA sequence (Table 1). The first-stand cDNAs of *AhLEC1A* were amplified using SYBR premix Ex Taq polymerase (Takara). Its relative expression level was analyzed using *AhACTIN7* as the reference gene by the $2^{-\Delta\Delta CT}$ method [30]. Three sample repetitions with technical triplicates were set in the experiment.

## In-silico analysis of the *AhLEC1A* promoter for cis-regulatory elements

The cis-elements of the 5' flanking region of *AhLEC1A* gene were analyzed using PLACE (http://www.dna.affrc.go.jp/PLACE) and Plant Cis-Acting Regulatory Elements (Plant CARE) (http://bioinformatics.psb.ugent.be/webtools/plantcare/html/).

## Plasmid construction and Arabidopsis transformation

A series of 5'-truncated promoter sequences were obtained by PCR using a single reverse primer localized in 5' UTR of *AhLEC1A*, and different forward primers situated in the different sites of the *AhLEC1A* promoter (The primer sequences were listed in Table 1). To construct the vector, the appropriate restriction sites were introduced into the PCR-amplified promoter (*HindIII* at the 5' end; *NcoI* at the 3' end). The PCR-amplified promoter was then inserted into *HindIII*/*NcoI*-digested pCAMBIA3301, replacing the cauliflower mosaic virus (CaMV) 35S promoter, producing seven deletion constructs containing various fragments (-2228 ~ +82, Q7; -1254 ~ +82, Q6; -935 ~ +82, Q5; -721 ~ +82, Q4; -617 ~ +82, Q3; -354 ~ +82, Q2; -105 ~ +82, Q1).

The constructs including Q1-Q7 and the control pCAMBIA3301 was introduced into *Agrobacterium tumefaciens* strain GV3101 using a freeze-thaw method. Transgenic Arabidopsis plants were generated by the floral dip method. The seeds of the $T_0$-$T_2$ generations were germinated on 1/2MS$_0$ agar medium containing 10μg/L Basta. The copy number in transgenic plants was determined by segregation ratio of the plants with and without basta-resistance. The $T_1$ transgenic lines with single copy gene have the 3:1 ratio of resistant plants to non-resistant plants. The homozygous lines of $T_2$ generation were screened on basta-resistant 1/2MS$_0$ medium. More than eight homozygous lines respectively carrying single copy gene of Q1-Q7 and the control pCAMBIA3301 were obtained. The identified transgenic plants were transferred to soil under 120 μmol·m$^{-2}$·s$^{-1}$light in a growth room at a temperature between 22˚C and 25˚C. All Arabidopsis plants grew under a 16h light/8h dark photoperiod, and 65% relative humidity.

### Histochemical GUS staining

The GUS assay was performed as described by Jefferson [31]. For each *AhLEC1A* promoter-GUS construct, at least thirty plants of $T_2$ generation lines in five transgenic events were used for GUS histochemical staining. The roots and leaves at the 4-leaf stage, stems at the bolting stage, flowers, immature embryos of 6–10 days after pollination and 3-5day etiolated and de-etiolated seedlings in transgenic $T_2$ lines were incubated in GUS assay buffer with 50mM sodium phosphate(7.0), 0.5mM $K_3Fe(CN)_6$, 0.5mM $K4Fe(CN)_6 \cdot 3H_2O$, 0.5% Triton X-100, and 1mM X-Gluc at 37°C overnight and then cleared with 70% ethanol. The samples were examined by stereomicroscopy.

## Results

### Isolation of the promoter of *AhLEC1A* and localization of TSS

The 2739bp DNA fragment was amplified by two rounds of PCR using the method of genome walking. Its sequence analysis found that this fragment includes 2707bp of 5' flanking region upstream from ATG and 32bp of coding sequence (Fig 1). In order to determine the

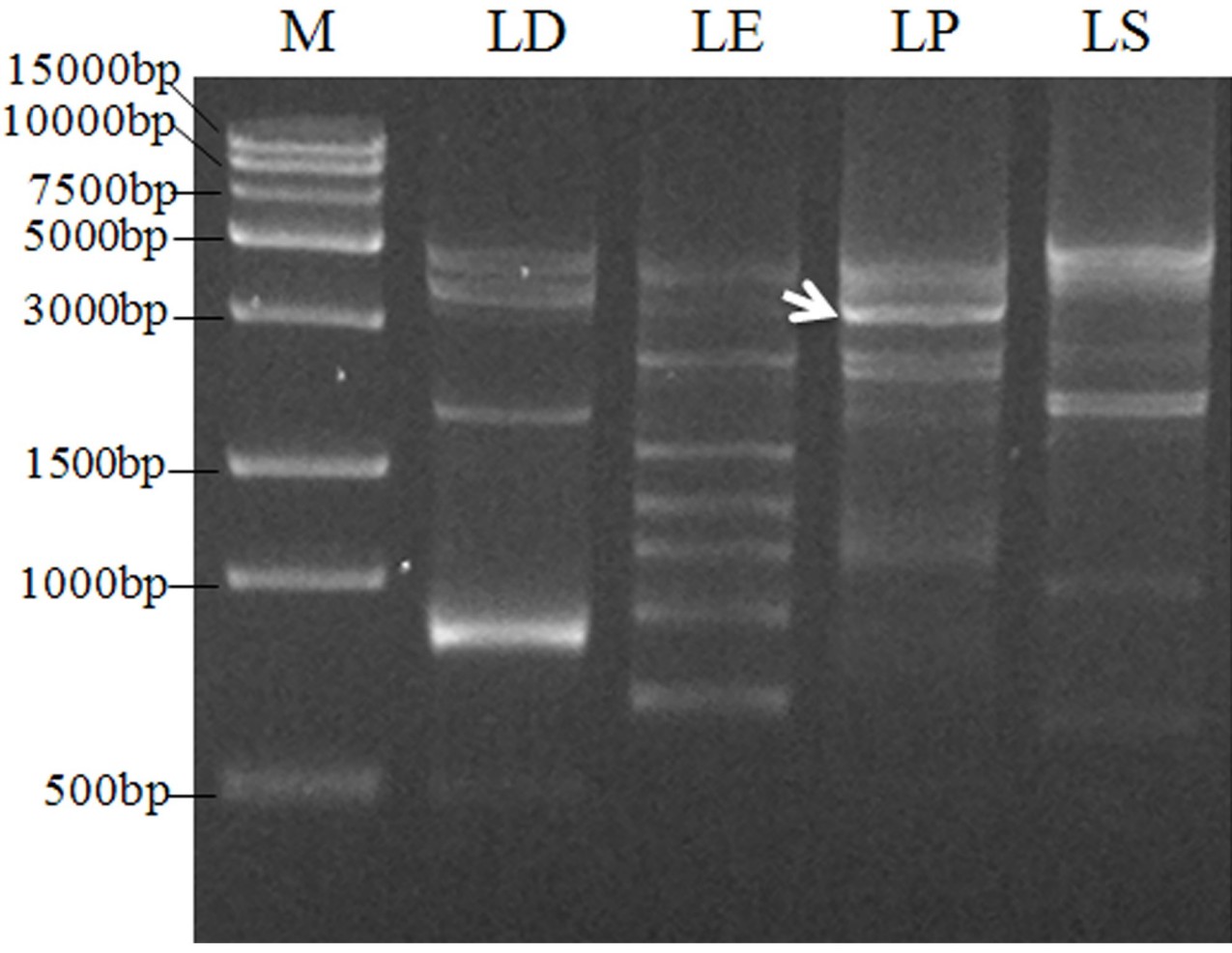

**Fig 1. PCR amplification of 5' flanking regulation regions of peanut *AhLEC1A* gene by chromosome walking.** LD, LE, LP and LS represents the second amplification with different primary product as template respectively. The arrow indicates the targeted band for further cloning and sequencing.

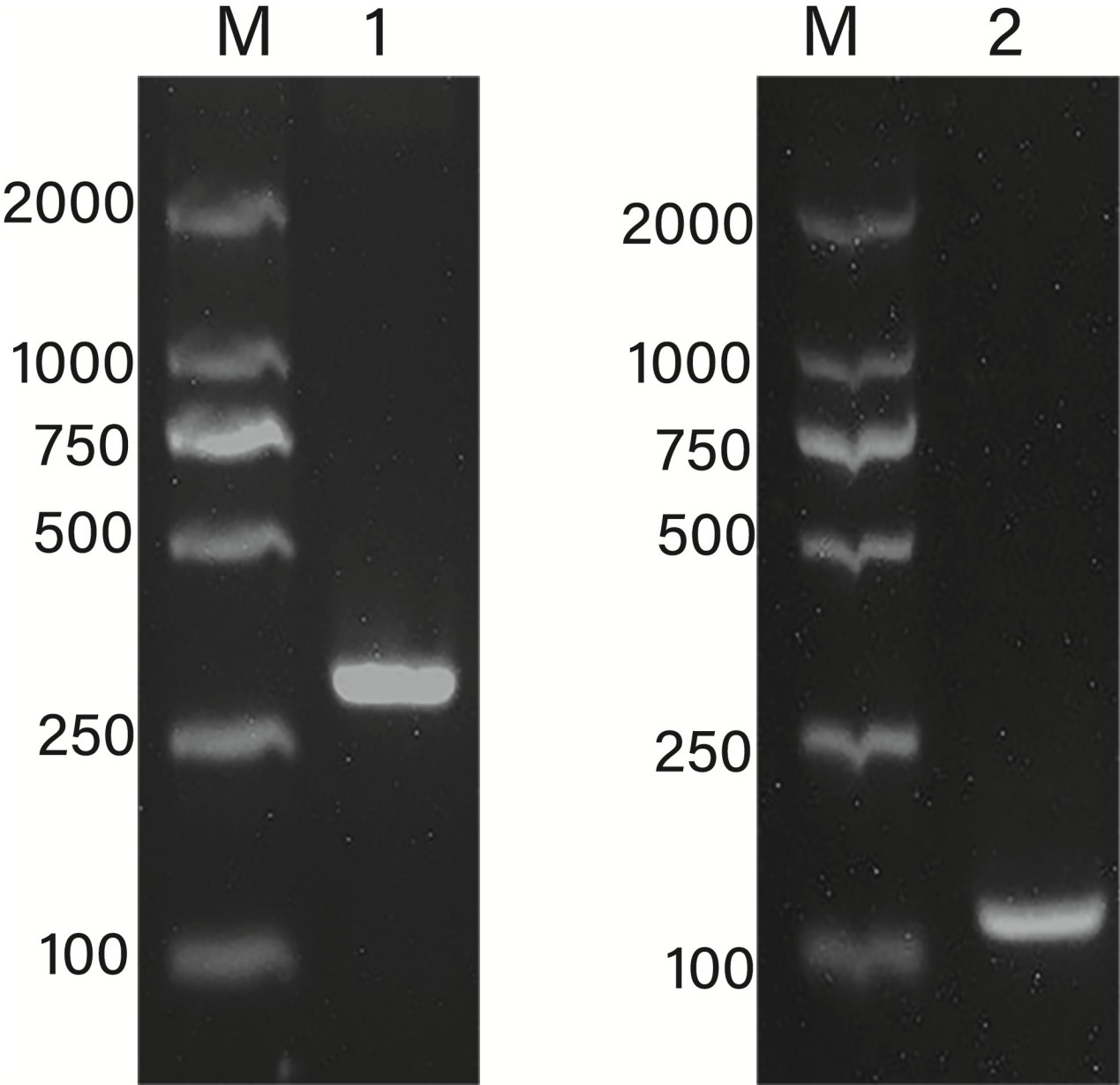

**Fig 2. Localization of transcription start site of peanut *AhLEC1A* gene using 5' RACE.** 1-The products of the first round PCR; 2-The products of the second round PCR.

transcription start site (TSS) of *AhLEC1A* gene, the nested 5' RACE was performed to amplify the 5'-end of its cDNA. The 140bp of cDNA fragment, including the 58bp of coding region started from ATG and 82bp 5' UTR, was isolated (Fig 2). Compared with the gDNA sequence of *AhLEC1A*, the sequence of 82bp 5' UTR was identical to the 5' upstream sequence of its gDNA, suggesting that the "A" located at the 82th nucleotid (nt) upstream from ATG is the TSS of *AhLEC1A* gene.

## Analysis of cis-regulatory elements in *AhLEC1A* promoter sequence

*In silico* analysis of 2707bp 5' flanking region revealed that a number of putative cis-elements were present in the 2625bp of promoter region and 82bp of 5' UTR (Fig 3). In detail, the basic

```
-2625  ATCTTTCTAAAAATAATTTTAAAAGAGAGAATAGCGCTTTAATTACATGAGAATAGTACTTTCTAAAAATAAAGAT
-2549  ATTTTTCTAAATTAGTATAATTATAATAAAGATATTTTTATAAAATAAACTTAGAAAATAAAATAGTGCATTTATTT
-2472  TGACGGAAAAATGGATTCATGAGGAATTGACACCTCATTTCCATTAACTTGGAAAAAACTCAATTTTAGTATATTA
       WBOXATNPP1               WBOXATNPP1
-2396  AGTAGATATATAATAAAAAGTAAAAAAAAAAACTTAAATAAAATTTTGATTTCTATAATCTTTTTAATTTTGGTTC
-2320  AATAAAATTTAAAATTTATATTTTAGTTTTGATCTATTAGTCATATGACAAAATTATGTTTGAAACCAAAATTAAAW
                                             RKY710S  WRKY710S/CARGCWBGAT
-2243  ATAAGATTTTATAAAGCTTAAAAAGTTGAACATTTTATATAGACCACAAATTTATTTAAGAAAAATAAAGTGTTTC
                            P8
-2167  TTAATTAACTATGACCATGAAGTGTCAATTGAAGGTGTATTTAGTTGAGTATTTGAGTTACAAGGCGTTCAATGGTT
                      WRKY710S
-2090  TGAGATGTCTCCTTTTTTTTGTTTAGCTTTTTTTTTTTTTCCTTTTAAAGACAAATGCAATTCTATATTTTAAGAAATT
                   CARGCWBGAT
-2011  GTAGTTTTTGCCATCAATAATTATAATTGAAAAAATAATCAAAAATTTTACTTTTTTTGTCAATGTTATCCTTCAGTC
                                                              WRKY710S
-1933  TTATATAATGTATATTGTTATCTTTTCTTTATGAAATTCTTAATACTCATTATTTATATAAAATTTCTAATATTTTTAC
-1854  TAATACAAATTTCTTTTTATGAATTTTTTTTGTTATTTTTTCGTTTGGTTTTGTATAATTCTTATCGTGAATTCTTGAA
-1775  TTAAAATGTACCCTGTTTATTGTGATTAATATTTTATAATATAAATTATTGATCTTAGAAAGGCCATATTAAATAAA
-1698  AAATATTAAAAAAATGAGTATTGAAAAGTTATTATTAAGAGTATTAAATGTCAGCACATAAATTTAAGTTCTTTTA
       ROOTMOTIFTAPOX1                                         WRKY710S
-1622  AAAAAATTATGACAAAGAAGTGTTTGAAAAAATGTGGCTTTGAATAATATAAATTCATATATTTTGTTAGTAATTT
                WRKY710S   CANBANAPA/2S SEED PROT BANAPA
-1546  TGATTTTAAATAATATAATTTAAATAACATTATTTTATTATAATTCATTTTTATATATAAATTACTAAACATATATCG
-1468  TATTAACACGCTTTTTTATTAAAATCAAGTTTGTAAAATCAATTTATATAAAAAAAATTTGTAAATTTTAATCTAAA
-1391  TACACACTAAACTTTGAGTTGATATTTAAATGGGTTACTGCCTATAATATTTAAACGTTCAACACTCGTGTAAATAA
           CACTFTPPCA1               SEF3 MOTIFGM                  DPBFCORE DCDC3
-1314  ATTAATGAATTATTAAACTGTATTTAAAATAAAAATTAAAACTGAGTGATTGATATTTAAAGATCGAAACTAATTT
                                                              P7        CARGCW8GAT
-1238  AAGTTTTAAATTTTTTATATTATATTTGATATAAAAATATATAAAATTAAATTATATTTCAGTATTTTATTTAGTTTAA
-1160  GATAAATATAAAAATAAAATAAAAATATTTATTAAAATATTTTTAATTATTAAAAAAAATATATTAAACATTTAGTT
       I BOX
-1084  TTTATTTTCAAAAATTTTAATATTTTGTAATCTCACTTTTTAAAAATATTAAAAAAACTAAAATTTTATATTTTTAAA
-1006  AAAAAATTATTTATAATCTTTACTTATCAAATACAATATTAAATCTCAATTTAAATTCCAATCTCTTAAAATACATG
                                                                          P6
-929   GCACGCCTCATATCTATTATCACTTTATCATAACTCAAAAGGATTTCCTGCCAATACTGGTCACTGGCCCACTGCAA
                                                                  WRKY710S
-852   TTTTATTACCCGTTATTGTAAGAGCATTACTTCTGTTCAAGTTTATCAGTTACACCTAAGTACATGTACTCTTGTAAT
                      TCT-motif
-774   TCCCAAGGAAAAAAAGTAAATTTATATCACTTGTATTTCAAAAACACCATCTTGTAATTTTTGGATAGCTTGTTGAA
                                                              P5
-697   GTAAATTTAACTGTATAGTGCGGCTCAAGTGATCTGTCTAATTTACAAAAATAAAAAAATAGACACTAATTAGAAT
-621   TAAACCCGTTAAAAAAAATAATAAGAATGTAACTACAAAAAATTAAAGTTAATAATTTTAAAGTCTCTAAATATA
       P4
-546   ATTATATTTTATTGGAATATTTTAATATATATTAGTTATCAAAAAACATTTTGTATCATTACTATTAATGACTCTGTA
                     ROOTMOTIFTAPOX1                                      SKn-1 motif
-468   AATAGTTATATATAAAACAAGTGTTTGAAATTATTCTTAAAATCACTAATTATATAAAAAAAAAGATAAATATTTAG
                        DPBFCORE DCDC3 /CANBANAPA//2S SEED PROT BANAPA        ROOTMOTIFTAPOX1
-392   TTATGATTTTTTTCTTCCACGATAATGATAAAATATTATCTTGGCAATAAATGTTGGGATTAATAAATTTTAGCAAT
                        I BOX              ATATT P3
                                          ROOT MOTIF TAPOX1
-315   TTTGTTAAATAATTAATCAGTATAAAAATAAAATTATTTTTAATAAATAAATTTAATTGATTTATATACATAAATTC
-238   TGATGAATATAAATATAAATATAAATATAAATGATTTTTGATTAAGTAGGAGCTTAAAATAATAACATTGGATAG
-162   CATACGGTAAAATAAATTTGGAATACTTGTGAAAAAAAAAAAACGTACAAGTGTGGGTGCATTAGGGTCAAAAGAGT
                       CAAT BOX                       DPBFCORE DCDC3  P2    WRKY710S
-87    GTAAATGACTGGTTATTGGCTGTTTCCTCGGTTGCATCGCCCCTTTGTTCACTATATATCTTGCCACTTGCCAACTACT
       SKn-1 motif/WRKY710S                                       TATA BOX
-10    TCATTGCCCAAACTTCACTTCATTTCATCATCACTGCTAACTAGATTAGCAGCTACTTACCTGCTCTCTCTCACTGCA
       transcription start site        CACTFTPPCA1                        CACTFTPPCA1
+70    TCTTTCTTCACACATGGAAACTGGAGGAGGCTTTCACGGTTATCA
               P1
```

**Fig 3. The sequence of 5' flanking regulation region of peanut *AhLEC1A* gene and some major elements harbored in this region.** The letter "A" in box represents its transcription start site (TSS), the putative regulatory elements are highlighted by underling or italicizing, and the sequences of primers P1-P8 are shading.

promoter elements, TATA box (TATATAT) and CAAT box (CAAAT)), respectively placed at -36 ~ -30 nt and -143 ~ -148 nt. Many crucial elements required for embryo- or endosperm-specific expression and seed storage compounds accumulation scattered over the promoter, including two SKn-1 motifs (GTCAT) at -84 ~ -80 nt and -475 ~ -479 nt, two CANBANAPA elements (CNAACAC) at -442 ~ -448 nt and -1597 ~ -1603 nt, and three binding sites for AGL15 (CWWWWWWWWG) at -1245 ~ -1236 nt, -2079 ~ -2070 nt and -2272 ~ -2263 nt. In addition, four DPBFCOREDCDC3 elements (ACACNNG), previously considered to involve in embryo-specific expression and also to respond to ABA were found at -111 ~ -117 nt, -445 ~ -451 nt, -1321 ~ -1327 nt and -1330 ~ -1324 nt. We have also detected many other regulatory elements for the accumulation of seed storage compounds and embryogenesis. For instance, eight EBOX BNNAPA (CANNTG), two 2S SEED PROT BANAPA (CAAACAC) and one SEF3 MOTIF GM (AACCCA). Besides, there were some elements associated with regulating in vegetative organ development on the promoter, such as mesophyll-specific element CACTFTPPCA1 (YACT), root-specific element ROOTMOTIFTAPOX1 (ATATT) and so on. There also exist some elements involved in light responsiveness including more than a dozen I BOX (GATAA) at -366 ~ -362 nt, -372 ~ -368 nt, -405 ~ -401 nt, -357 ~ -353 nt, -511 ~ -507 nt, -810 ~ -806 nt, -905 ~ -901 nt, -913 ~ -909 nt, -983 ~ -979 nt, -1793 ~ -1789 nt, -1916 ~ -1912 nt, and -1946 ~ -1942 nt, three -10PEHVPSBD (TATTCT) at -437 ~ -432 nt, -2584 ~ -2579 nt, and -2606 ~ -2601 nt, and one TCT-motif (TCTTAC) at -830 ~ -835 nt, etc., and some other regulatory elements controlling the chloroplast genes expression, like one GT1 MOTIF PSRBCS (KWGTGRWAAWRW) at -137 ~ -126 nt, and two etiolation-induced expression elements ACGT ATERD1 (ACGT) at -122 ~ -119 nt, and -1334 ~ -1337 nt. We also identified thirteen negative regulatory elements in the promoter, among which five WBOX-ATNPR1 elements located in -1255 ~ -2228 nt region (-1613 ~ -1616 nt, -1649 ~ -1652 nt, -1953 ~ -1956 nt, -2144 ~ -2147 nt, and -2156 ~ -2159 nt), and four WRKY71OS elements densely distributed between the region of -2228 to -2625 nt.

## Functional analysis of the regulatory regions of the *AhLEC1A* promoter

To validate the role of the crucial regulatory region in *AhLEC1A* promoter, a series of GUS expression vectors (Q1~ Q7) (Fig 4), driven by different length of the promoters with truncated 5' terminal were established, and the GUS expression patterns in stable transgenic plants of *Arabidopsis* was investigated. In the histochemical assay, GUS expression was visualized specifically in the developing embryos of transgenic plants containing Q7 construct (including 2228bp promoter region and 82bp 5' UTR, Fig 5). The result of *AhLEC1A* expression analysis by qRT-PCR also showed that its transcripts were higher in seeds, but lower or rarely in roots, stems, leaves and flowers (S1 Fig) Otherwise, the GUS staining was observed in all detected organs of transgenic plants carrying Q3, Q4, Q5, and Q6 construct (Fig 5). These four promoter segments are respectively 617bp, 721bp, 935bp and 1254bp in size with 5' terminal deletion of 1611bp ~ 974bp. It was suggested that there exist some key motifs in the promoter region between -2228bp and -1255bp, which related to inhibit the expression in the other organs except for the developing seeds. Moreover, the further deletional promoter fragment Q2 with 354bp drove the GUS to express only in embryos and rosette leaves. The shortest fragment Q1 containing 105bp promoter region and 82bp 5' UTR couldn't drive the GUS to express in any detected organs of transgenic *Arabidopsis* (Fig 5), implying that it might be caused by the deletion of the necessary component for gene expression.

To explore the role of *AhLEC1A* on seedling establishment, the transgenic lines with Q7, Q5, Q3 and Q2 constructs were chosen for further analysis. Transgenic Arabidopsis seeds were kept in the dark till their germinating. The results of GUS staining indicated that the

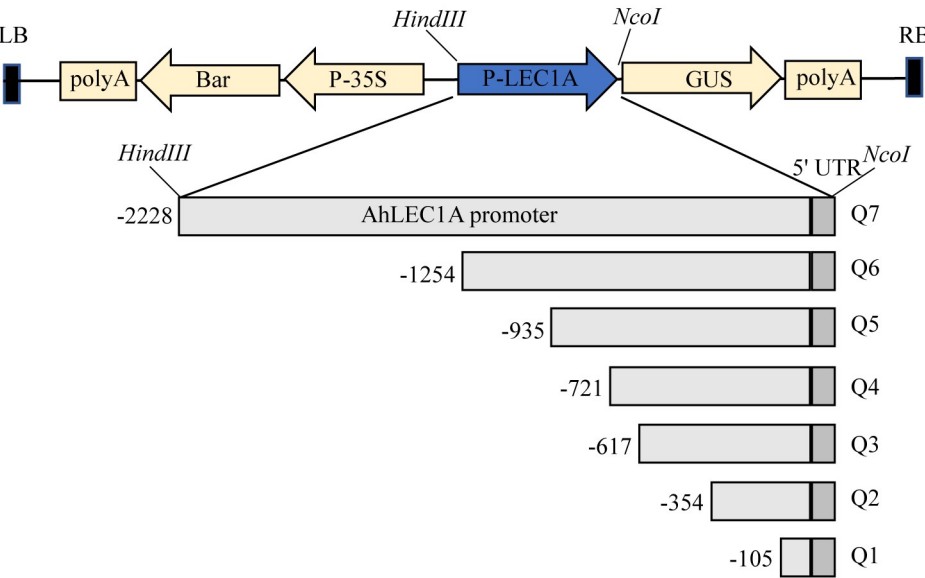

**Fig 4. The vector diagram expressing GUS in plants driven by different length *AhLEC1A* promoters with 5' terminal deletion.** Q1~Q7 indicates its promoters with different length. The rectangles in light and dark gray respectively show the promoter region upstream of the TSS, and the 5' UTR region of gene.

unexpanded cotyledon and apex hook of the seedlings harboring Q7 or Q5 construct showed dark blue color, and the hypocotyls were light blue. However, after the etiolated transgenic seedlings had been moved to the light for 2 days, the plants with Q7 construct hardly got dyed, and only the expanded cotyledons with Q5 construct were stained blue. By contrast, the whole seedlings with Q3 or Q2 construct were dyed dark blue under both growth conditions (Fig 6). The results suggested that there existed some negatively regulatory elements at the region of -2228bp ~ -618bp in *AhLEC1A* promoter to control the expression of *AhLEC1A* in hypocotyls

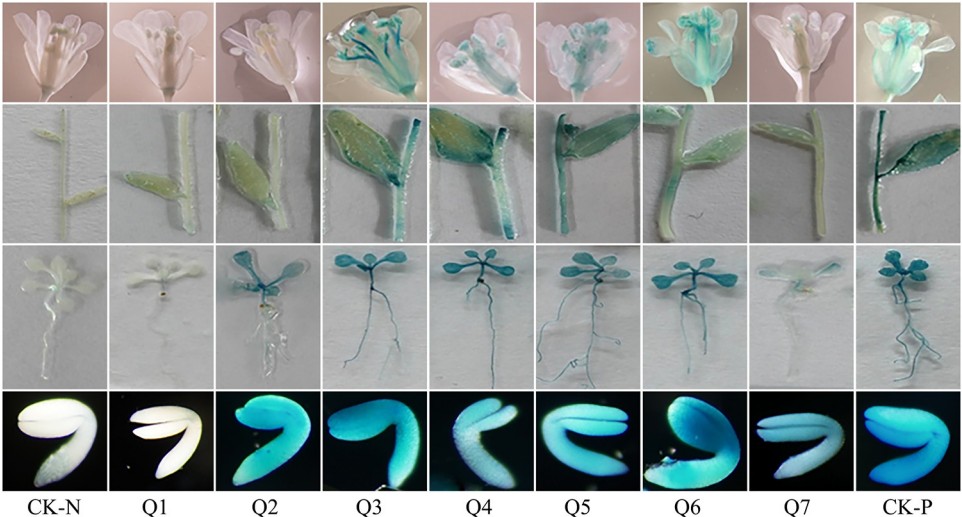

**Fig 5. GUS Histochemical staining of different organs in transgenic Arabidopsis.** Q1~Q7 respectively shows the GUS expression patterns in flower, stem and cauline leaf, and rosette leaf and root harboring different GUS expression structures, and the CK-N and CK-P respectively show the GUS expression profiles in COL, and in positive control harboring 35S::GUS constructs.

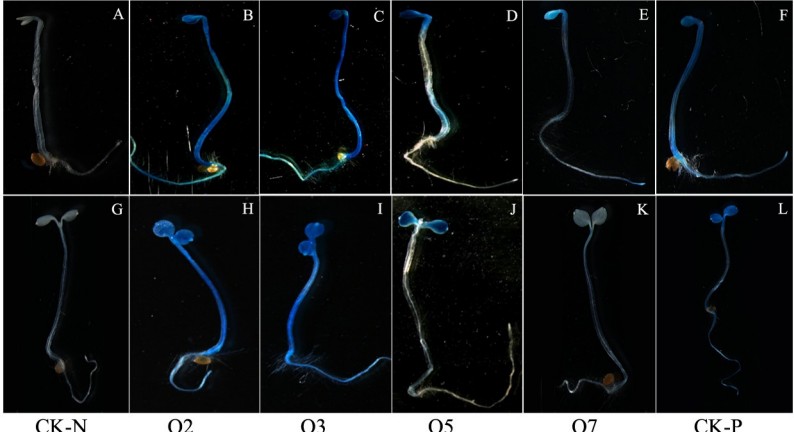

CK-N          Q2          Q3          Q5          Q7          CK-P

**Fig 6. GUS staining of the seedlings germinated for 3~5d in transgenic Arabidopsis.** Q2, Q3, Q5 and Q7 indicate respectively the GUS expression patterns in transgenic Arabidopsis lines harboring different GUS expression structures. CK-N and CK-P indicate the GUS expression profiles in COL and in positive control harboring 35S::GUS construct, respectively. A-F indicate the seedlings germinated for 3d in dark, and G-L represent the seedlings after transferring from dark to light for 2 days.

and radicles at the stage of seedling formation, and some of them mentioned above might associate with light response.

## Discussion

Identifying and characterizing the 5' flanking region of gene is helpful for revealing its temporal and spatial expression pattern, and facilitating its utilization in plant genetic engineering [32]. In the present study, we have cloned and analyzed the 5' flanking regulatory sequence of *AhLEC1A*. Several cis-elements in *AhLEC1A* promoter, such as SKn-1, CANBANAPA $(CA)_n$, AGL15, DOF core, SEF3 motif and the like, which previously were demonstrated to be required for seed development and storage accumulation, were identified. Skn-1 motif and $(CA)_n$ element were reported to play a vital role in determining the seed-specific expression; the deletion of Skn-1 motif or $(CA)_n$ element in *glutelin* and *napin* promoter decreased their transcription in seeds [33, 34]. The element of DOF core was considered to confer the endosperm-specific expression in *Zea mays* [35, 36]. SEF3 motif is the binding site of Soybean Embryo Factor 3, which regulate the transcription of the β-conglycinin (a storage protein) gene and participate in seed development [37, 38]. Our results of GUS staining assay revealed *GUS* gene, driven by the longest *AhLEC1A* promoter (Q7), specially expressed in the embryos of the transgenic Arabidopsis, which is well in agreement with our results of gene expression analyzed by qRT-PCR method (S1 Fig). These data showed that AhLEC1A functioned in a seed-specific manner. Otherwise, the transgenic lines with 1611bp ~ 974bp deletion constructs from 5' terminal of Q7 promoter showed the constitutive expression at higher GUS levels in roots, rosettes, stems, flowers, and seeds. Meanwhile, in silico analysis of *AhLEC1A* promoter displayed several tissue-specific elements like mesophyll-specific element CACTFTPPCA1, root-specific element ROOTMOTIFTAPOX1 and pollen-specific element POLLEN1LELAT52 distributed on its upstream regulatory region, as well as many negatively regulatory elements including four WRKY71OS and five WBOXATNPR1 dispersed intensively in the fragment of -1225 ~ -2228bp which was deleted in Q3 ~ Q6. These results demonstrate that *AhLEC1A* expression in seed-specific pattern might be attributed to be negatively regulated its transcription in vegetative organs by some cis-element existed in the distal region of its promoter, and

simultaneously to be controlled its expression in seeds positively by some seed-specific elements in the proximal region of its promoter. This regulatory model was also found in *AtLEC1* promoter of Arabidopsis [27], *D540* promoter of rice [39], and *C-hordein* promoter of barley [40].

Beyond embryogenesis and embryo development, LEC1 also regulate skotomorphogenesis of seedlings at the post-germination stage. The unexpanded cotyledons and apical hook of seedlings with Q7 construct germinated in dark dyed obviously in blue, while the whole seedlings were scarely stained after transferring to light for 2 days. It was suggested that light might repress the expression of *AhLEC1A* by recruiting some proteins to bind the particular elements in its promoter. Our results found that total 23 light-responsive elements I BOX CORE/GATA BOX (GATA) scattered on Q7 segment of *AhLEC1A* promoter, 9, 15 and 21 out of them were respectively deleted in Q5, Q3 and Q2 promoter, resulting in that in GUS assay, the staining patterns of Q2 and Q3 transgenic plants in darkness were similar to those in light, and the hypocotyls of Q5 transgenic plants were dyed in blue when growing in darkness while there were no dyeing after transferring them to light. The core sequence of I BOX, and the GATA BOX with similar function had been shown to be essential for light-regulated transcriptional activation [41–43]. Furthermore, it has been demonstrated that I BOX as a negative cis-element can inhibit the expression of *GalUR* in strawberry, and the inhibited role is strictly depended on light [44]. Yamagata et al. also found that I BOX, as a negative regulatory element, was necessary for down-regulating the expression of *cucumisin* gene by binding fruit nuclear protein in Musc melons (*Cucumis melo* L.) [45]. Previous study found that *AtLEC1* promoter exists several I BOX CORE elements, and deleting some of them localized on 5' upstream segment from -436 nt in mutant *tnp* restrains the hypocotyl elongation of etiolated seedlings in darkness [27]. These data suggested that some of I BOX elements function as a negative regulator in response to illumination. In our GUS histochemical assay, the degree and range of dyeing changed with the number of I BOX, demonstrating that some of them might be involved in negative regulating the expression of *AhLEC1A* gene during the procedure of seedling growth from dark condition to light condition.

The *cis-* elements comparison in the promoters of *AhLEC1A* and *AhLEC1B* showed that lots of similar elements are dispersed in the both promoters, but their amounts and positions were much different (Table 2). *AhLEC1A* promoter contained a number of distinct seed-development related components such as 2S SEED PROT BANAPA, SP8BFIBSP8BIB, CANBNNAPA. However, *AhLEC1B* promoter contained numerous specific elements involved in abiotic stress or hormones responding, including GCCCORE, ASF1MOTIFCAWV, and several regulatory elements known to modulate gene expression at higher transcription level in different plant species [46]. The similarities and differences between two *AhLEC1* promoters implied that their functions might be partially same and redundant, and to some extent *AhLEC1A* and *AhLEC1B* might play different roles during the particular growth and development period of peanuts, respectively. The point of view was consistent with the study of predecessors who thought *LEC1* genes originated from a common ancestor and neofunctionalization and /or subfunctionalization processes were responsible for the emergence of a different role for *LEC1* genes in seeds plants [47]. Moreover, during the evolution of cultivated peanuts, A and B subgenomes were subjected to asymmetric homoeologous exchanges and homoeolog expression bias. Yin et al. considered that A subgenome were significantly affected by domestication, while natural selection preferred to B subgenome [48]. It was speculated that during genome evolution, to satisfy the demands for seed growth and development, the orthologous genes *AhLEC1A* and *AhLEC1B* suffered from the different selection pressure at different life stages to produce their functional divergence.

**Table 2. Comparison of regulatory elements in *AhLEC1A* promoter and *AhLEC1B* promoter.**

| *cis*-element | *AhLEC1A* promoter | *AhLEC1B* promoter | Motif[a] | Putative function |
|---|---|---|---|---|
| Skn-1 motif | + | + | GTCAT | Cis-acting regulatory element required for endosperm expression [49] |
| CARGCW8GAT | + | + | CWWWWWWWWG | Motif with a longer A/T-rich core providing binding site for AGL15 which accumulates during embryo development [50] |
| CACTFT PPCA1 | + | + | YACT | Cis-acting regulatory element required for mesophyll-specific expression [51] |
| -10PEHVPSBD | + | + | TATTCT | Cis-acting regulatory element involved in light responsiveness [52] |
| ROOT MOTIF TAPOX1 | + | + | ATATT | Motif found in the promoter of rolD, which expresses strongly in roots [53] |
| POLLEN1 LELAT52 | + | + | AGAAA | A regulatory element responsible for pollen specific activation of gene [54] |
| WRKY71OS | + | + | TGAC | Binding site of rice WRKY71, a transcriptional repressor of the gibberellin signaling pathway [55] |
| W BOX ATNPR1 | + | + | TTGAC | A cluster of WRKY binding sites act as negative A regulatory element for the inducible expression of genes [56] |
| CPB CSPOR | + | + | TATTAG | Cis-Acting regulatory element involved in Cytokinin responsiveness [57] |
| DOF CORE ZM | + | + | AAAG | Core site required for binding of Dof proteins which may be associated with the plant-specific pathway for carbon metabolism [35] |
| E BOX BNNAPA | + | + | CANNTG | The cis-elements in the promoter regions of most genes encoding the storage protein [58] |
| ERE LEE4 | + | + | AWTTCAAA | The ethylene responsive element mediate ethylene-induced activity of transcription [59] |
| SEF3 MOTIF GM | + | + | AACCCA | Binding with SEF3, one of soybean embryo factor (SEF) [37] |
| I BOX | + | + | GATAA | Conserved sequence upstream of light-regulated genes [41] |
| ARFAT | + | - | TGTCTC | Cis-Acting regulatory element involved in auxin responsiveness [60] |
| CAN BANAPA | + | - | CNAACAC | Core of (CA)n element in storage protein genes [32] |
| 2S SEED PROT BANAPA | + | - | CAAACAC | Cis-Regulatory element conserved in many storage-protein gene promoters [58] |
| ATC-motif | + | - | AGCTATCCA | part of a conserved DNA module involved in light responsiveness [61] |
| ERE | + | - | ATTTCAAA | ethylene-responsive element [62] |
| GARE AT | + | - | TAACAAR | gibberellin-responsive element [63] |
| SP8BFIBSP8BIB | + | - | TACTATT | "SP8b" found in the 5' upstream region of three different genes coding for sporamin and beta-amylase [64] |
| TCT motif | + | - | TCTTAC | part of a light responsive element [65] |
| ASF1MOTIFCAMV | - | + | TGACG | Motif involved in transcriptional activation of genes by auxin or salicylic acid, may be relevant to light regulation [66] |
| GCCCORE | - | + | GCCGCC | Core of GCC-box found in pathogen-responsive, ethylene-responsive and jasmonate-responsive gene [67] |
| TGACGTVMAMY | - | + | TGACGT | Motif required for high level expression in cotyledons of the germinated seeds [68] |
| 5'UTR Py-rich strech | - | + | TTTCTTCTCT | Cis-acting element conferring high transcription levels [69] |
| CGTCA-motif | - | + | CGTCA | Cis-acting regulatory element involved in the MeJA- responsiveness [70] |
| GAG-motif | - | + | AGAGAGT | Part of a light responsive element [71] |
| GARE-motif | - | + | AAACAGA | Gibberellin-responsive element [72] |
| LTRE1HVBLT49 | - | + | CCGAAA | "LTRE-1" (low-temperature- responsive element) in barley (H.v.) blt4.9 gene promoter [65] |

Note: "+" means the element existing in the promoter, "-" means the element not existing in the promoter.

[a]W = A/T; Y = T/C; N = G/C/A/T.

In summary, we identified and characterized the promoter of *AhLEC1A*. It was found that during the process of seed development and maturation, its expression in embryo were regulated by the positive cis-elements in seed-specific mode and the negative elements restricting its expression in other organs. Moreover, *AhLEC1A* was also involved in skotomorphogenesis

of peanut seeds, its expression level in the hypocotyls germinated in darkness was inhibited by some light-responsive elements. The results will be helpful for understanding the function of *AhLEC1A* in peanuts.

## Supporting information

**S1 Fig. Expression patterns of *AhLEC1A* in different organs.** The transcription levels of *AhLEC1A* mRNA in various organs were analyzed by qRT-PCR with *AhACTIN* 7 as internal referent gene. R: Roots; St: Stems; L: Leaves; F: Flowers; S: Seeds after pegging for 30 d. (TIF)

**S1 Raw images.**
(PDF)

## Author Contributions

**Data curation:** Pingli Xu, Guangxia Chen.

**Formal analysis:** Guiying Tang, Pingli Xu, Pengxiang Li.

**Funding acquisition:** Lei Shan, Shubo Wan.

**Methodology:** Jieqiong Zhu, Guangxia Chen.

**Resources:** Jieqiong Zhu.

**Software:** Pengxiang Li.

**Writing – original draft:** Guiying Tang.

**Writing – review & editing:** Lei Shan, Shubo Wan.

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
