## [Decision Letter · Decision Letter 0]

21 Dec 2020

PONE-D-20-35417

Cloning and Functional characterization of seed-specific LEC1A promoter from peanut (Arachis hypogaea L.)

PLOS ONE

Dear Dr. Shan,

Thank you for submitting your manuscript to PLOS ONE. After careful consideration, we feel that it has merit but does not fully meet PLOS ONE’s publication criteria as it currently stands. Therefore, we invite you to submit a revised version of the manuscript that addresses the points raised during the review process.

We look forward to receiving your revised manuscript.

Kind regards,

Keqiang Wu, Ph.D

Academic Editor

PLOS ONE

Journal Requirements:

2.Thank you for stating the following financial disclosure:

 "No"

3.Thank you for stating the following in your Competing Interests section: 

 "No"

Reviewers' comments:

Reviewer's Responses to Questions

**Comments to the Author**

1. Is the manuscript technically sound, and do the data support the conclusions?

Reviewer #1: Partly

Reviewer #2: Yes

2. Has the statistical analysis been performed appropriately and rigorously? 

Reviewer #1: N/A

Reviewer #2: I Don't Know

3. Have the authors made all data underlying the findings in their manuscript fully available?

Reviewer #1: Yes

Reviewer #2: Yes

4. Is the manuscript presented in an intelligible fashion and written in standard English?

Reviewer #1: Yes

Reviewer #2: Yes

5. Review Comments to the Author

Reviewer #1: Overall, this is a good experimental design. Here are some comments for the authors’ consideration -

1. The identified AhLEC1A is a homolog of Arabidopsis LEC1 – has this been supported by sequence alignment in this report or supported by appropriate references?

2. It is a logical alternative to use Arabidopsis for the promoter:GUS analysis, given that peanut was a difficult option to produce transgenic plants. However, the promoter;GUS analysis could be corroborated by tissue-specific RT-PCR to confirm the presence of the respective mRNA. Or, at least the expression of the similar Arabidopsis LEC1 promoter:GUS deletion constructs could be used as a control for a comparative expression analysis of the AhLEC1A promoter components. In absence of either of these supporting studies, the author needs to acknowledge the shortcomings of the promoter:GUS analysis alone in claiming spatial and temporal expression of AhLEC1A. (Ref #35 that claims to have shown expression of AhLEC1 is not available for non-Chinese readers and also it seems to support the Q7 GUS expression only)

Reviewer #2: In the current study, the authors have cloned a 2707 bp upstream sequence of the AhLEC1A gene from peanut and confirmed its transcription start site by 5' RACE. In addition, this study dissected the functions of the different regions of the AhLEC1A promoter under normal and dark/light conditions by using a promoter::GUS transgene approach in Arabidopsis. GUS staining showed that the 2300-bp AhLEC1A promoter (containing 82 bp of 5' UTR and 2228 bp promoter) was a seed-specific promoter and the expression pattern of this promoter may be controlled by several positive seed-specific regulatory elements and other negative regulatory elements. These findings will enhance the understanding of the regulatory mechanisms of the 2300-bp AhLEC1A promoter. I recommend it for publication in the journal after minor revisions.

Some points for the authors to consider in revision,

1. Research on the expression pattern of the LEC1 genes or functional analysis of the LEC1 promoters in different plants should be added in introduction. This will help readers understand the progress of the LEC1 promoter research.

2. The description of the transgenic lines is not detailed enough. The authors have not analyzed whether the transgenic Arabidopsis lines are single-copy or not. The expression and regulation of the transgene is correlated with the copy number of inserted transgenes.

3. The number of the transgenic lines in GUS staining analysis is a little small. Are the expression patterns of most of these transgenic lines (the major expression of the transformants) the same or similar?

4. The authors have cloned a 2707-bp upstream sequence of the AhLEC1A gene from peanut. However, they have not analyzed the activity of this longer promoter. The shorter 2300-bp AhLEC1A promoter (containing 82 bp of 5' UTR and 2228 bp promoter), not the longer 2707-bp promoter, was identified as the full length promoter of the AhLEC1A gene (Page 18, line 246, full-length AhLEC1A promoter (Q7)). Why?

5. The authors are suggested to add the positions of the deleted promoters in Figure 3, which would be more convenient to understand the paper.

6. PLOS authors have the option to publish the peer review history of their article (what does this mean?). If published, this will include your full peer review and any attached files.

Reviewer #1: No

Reviewer #2: No

---

## [Author Response · Author response to Decision Letter 0]

18 Feb 2021

Dear editor and reviewers,

I am very grateful to reviwer's valuable comments and suggestions. I have modified our manuscript, and the details revised are explained as follows.

Reviewer #1:

Q1: 

The identified AhLEC1A is a homolog of Arabidopsis LEC1 – has this been supported by sequence alignment in this report or supported by appropriate references? 

Answer: We identified AhLEC1A as a homolog of Arabidopsis AtLEC1 by clustering analysis and sequence alignment. LEC1 belongs to the B-subunit members of Nuclear Factor of the Y box (NF-Y) subfamily. We identified 19 NF-YB proteins from peanut genome, and aligned them with their homologous proteins from Arabidopsis and soybean. Their phylogenetic relationships were shown as following figure. It found that AhLEC1A and AhLEC1B were clustered together with Arabidopsis NF-YB9 (AtLEC1) and NF-YB6 (AtLEC1-like). On the other hand, the blast of sequence revealed that the identified AhLEC1A protein have higher sequence similarities with AtLEC1 in Arabidopsis. AhLEC1A shares respectively 57.64% and 85.71% identities of amino acid with AtLEC1 in ORF and B domain. 

Q2: It is a logical alternative to use Arabidopsis for the promoter: GUS analysis, given that peanut was a difficult option to produce transgenic plants. However, the promoter;GUS analysis could be corroborated by tissue-specific RT-PCR to confirm the presence of the respective mRNA. Or, at least the expression of the similar Arabidopsis LEC1 promoter:GUS deletion constructs could be used as a control for a comparative expression analysis of the AhLEC1A promoter components. In absence of either of these supporting studies, the author needs to acknowledge the shortcomings of the promoter:GUS analysis alone in claiming spatial and temporal expression of AhLEC1A. (Ref #35 that claims to have shown expression of AhLEC1 is not available for non-Chinese readers and also it seems to support the Q7 GUS expression only)

Answer:

I agree with your comments and suggestions. The expressions of AhLEC1A in different tissues were analyzed by qRT-PCR, and showed to have the seed-specific pattern. We have added the relevant descriptions in the Materials and Methods(line 84-85, and line 118-123), and the Results section (line 218-220, S1 Fig.). And I have deleted the reference: “ Li AQ, Xia H, Wang XJ, Li CS, Zhao CZ, Bi YP. Cloning and expression analysis of peanut (Arachis hypogaea L.) LEC1. Acta Bot Boreal-OccidentSin. 2009; 29(9)(1730-1735. doi:10.1007/978-1-4020-9623-5_5 (In Chinese)”

Reviewer #2

Q1: Research on the expression pattern of the LEC1 genes or functional analysis of the LEC1 promoters in different plants should be added in introduction. This will help readers understand the progress of the LEC1 promoter research.

Answer:

I have supplied the relevant descriptions about the expression pattern of the LEC1 genes in different plants in line 50-51 and line 60-62, and cited three references (11, 21, 22) in the text. 

11. Calvenzani V, Testoni B, Gusmaroli G, Lorenzo M, Gnesutta N, Petroni K, et al. Interactions and CCAAT-binding of Arabidopsis thaliana NF-Y subunits. PLoS One. 2012; 7(8): e42902. doi: 10.1371/journal.pone.0042902. PMID: 22912760

21. Baud S, Kelemen Z, Thévenin J , Boulard C, Blanchet S, To A, et al. Deciphering the molecular mechanisms underpinning the transcriptional control of gene expression by master transcriptional regulators in Arabidopsis seed. Plant Physiol. 2016;171(6): 1099–1112. doi;10.1104/pp.16.00034 PMID: 27208266

22. Jo L, Pelletier JM, Hsu SW, Baden R, Goldberg RB, Harada JJ. Combinatorial interactions of the LEC1 transcription factor specify diverse developmental programs during soybean seed development. Proc Natl Acad Sci U S A. 2020; 117 (2): 1223-1232.  doi:10.1073/pnas.1918441117 PMID: 31892538

Q2: The description of the transgenic lines is not detailed enough. The authors have not analyzed whether the transgenic Arabidopsis lines are single-copy or not. 

Answer:

Many thanks for your comments. In fact, we analyzed the GUS expression driven by AhLEC1 promoters with different length using single-copy transgenic lines in current study. However, it was regretted to ignore the relevant descriptions. We have supplied the method of copy number analysis in transgenic lines in line 142-147 of the Materials and Methods section in revised manuscript. 

Q3:

The number of the transgenic lines in GUS staining analysis is a little small. Are the expression patterns of most of these transgenic lines (the major expression of the transformants) the same or similar?

Answer:

Sorry, we didn’t express clearly about the number of the transgenic lines in GUS staining analysis in original manuscript. In our study, at least thirty plants of T2 generation lines in 5 transformed events of each GUS construct were used for GUS histochemical staining. The amounts of transgenic plants are enough to analyze the GUS expression. We have made it clear in revised manuscript at line 152-153. In addition, most of plants in each transformed construct have the similar expression pattern, and some lines with basically consistent expression patterns were selected to be their representatives of GUS expression patterns in different organs or developmental stages.

Q4:

The authors have cloned a 2707-bp upstream sequence of the AhLEC1A gene from peanut. However, they have not analyzed the activity of this longer promoter. The shorter 2300-bp AhLEC1A promoter (containing 82 bp of 5' UTR and 2228 bp promoter), not the longer 2707-bp promoter, was identified as the full length promoter of the AhLEC1A gene (Page 18, line 246, full-length AhLEC1A promoter (Q7)). Why? 

Answer:

We have cloned a 2707bp upstream sequence from the ATG initiation codon site of AhLEC1A. The bioinformatics analysis revealed that most of elements, including some elements involved in embyogenesis and seed storage compounds accumulation, located in the region of -2228 ~ +82 nt (2300bp), and fewer elements distributed in -2229 ~ -2707 nt region. It suggests that the 2300bp promoter could mainly be responsible for regulating the spatiotemporal expression of AhLEC1A. Therefore, we analyzed the activity of the 2300-bp promoter, which was the longest one among all promoters analyzed in our study, instead of full-length 2707-bp promoter. Because it is not suitable for calling 2300-bp promoter as full length, we have corrected it in the text. 

Q5:

The authors are suggested to add the positions of the deleted promoters in Figure 3, which would be more convenient to understand the paper.

Answer:

According to your suggestion, we have marked the positions of primers P1-P8 using the shaded sequences in Figure 3. The P2-P8 as the forward primers (marked with ‘→’) together with reverse primer P1 (marked with ‘←’) respectively were used to amplify seven promoter segments Q1-Q7. Therefore, the deleted fragments at 5'-terminus of these promoters can be convenient to distinguish in Figure 3, and the lengths of the promoters have been clearly displayed in Figure 4.

---

## [Decision Letter · Decision Letter 1]

5 Mar 2021

Cloning and Functional characterization of seed-specific LEC1A promoter from peanut (Arachis hypogaea L.)

PONE-D-20-35417R1

Dear Dr. Shan,

We’re pleased to inform you that your manuscript has been judged scientifically suitable for publication and will be formally accepted for publication once it meets all outstanding technical requirements.

Kind regards,

Keqiang Wu, Ph.D

Academic Editor

PLOS ONE

Additional Editor Comments (optional):

Reviewers' comments:

Reviewer's Responses to Questions

**Comments to the Author**

1. If the authors have adequately addressed your comments raised in a previous round of review and you feel that this manuscript is now acceptable for publication, you may indicate that here to bypass the “Comments to the Author” section, enter your conflict of interest statement in the “Confidential to Editor” section, and submit your "Accept" recommendation.

Reviewer #1: All comments have been addressed

Reviewer #2: All comments have been addressed

2. Is the manuscript technically sound, and do the data support the conclusions?

Reviewer #1: Partly

Reviewer #2: Yes

3. Has the statistical analysis been performed appropriately and rigorously? 

Reviewer #1: Yes

Reviewer #2: Yes

4. Have the authors made all data underlying the findings in their manuscript fully available?

Reviewer #1: Yes

Reviewer #2: Yes

5. Is the manuscript presented in an intelligible fashion and written in standard English?

Reviewer #1: Yes

Reviewer #2: Yes

6. Review Comments to the Author

Reviewer #1: Dear Authors: thanks for addressing the comments/ feedback. The phylogenetic analysis and the qRT-PCR figure could be a part of the main manuscript. Good work.

Reviewer #2: (No Response)

7. PLOS authors have the option to publish the peer review history of their article (what does this mean?). If published, this will include your full peer review and any attached files.

Reviewer #1: No

Reviewer #2: No

---

## [Editor Report · Acceptance letter]

11 Mar 2021

PONE-D-20-35417R1 

Cloning and Functional characterization of seed-specific *LEC1A* promoter from peanut (*Arachis hypogaea* L.) 

Dear Dr. Shan:

I'm pleased to inform you that your manuscript has been deemed suitable for publication in PLOS ONE. Congratulations! Your manuscript is now with our production department. 

Kind regards, 

on behalf of

Professor Keqiang Wu 

Academic Editor

PLOS ONE